# Ca and Mg Concentrations in Spices and Growth of Commonly Sporulated and Non-Sporulated Food-Borne Microorganisms According to Marketing Systems

**DOI:** 10.3390/foods10051122

**Published:** 2021-05-19

**Authors:** José María García-Galdeano, Marina Villalón-Mir, José Medina-Martínez, Sofía María Fonseca-Moor-Davie, Jessandra Gabriela Zamora-Bustillos, Lydia María Vázquez-Foronda, Ahmad Agil, Miguel Navarro-Alarcón

**Affiliations:** 1Department of Nutrition and Bromatology, School of Pharmacy, University of Granada, 18071 Granada, Spain; info@farmaciagaldeano.com (J.M.G.-G.); marinavi@ugr.es (M.V.-M.); pepemm97@correo.ugr.es (J.M.-M.); sofiafonseca@correo.ugr.es (S.M.F.-M.-D.); j.gaby_@hotmail.com (J.G.Z.-B.); lydiavazfor8@correo.ugr.es (L.M.V.-F.); 2Nutrition and Food Technology Institute of Granada, University of Granada, 18100 Granada, Spain; 3Department of Pharmacology, and Neurosciences Institute, School of Medicine, University of Granada, 18012 Granada, Spain; aagil@ugr.es

**Keywords:** Ca and Mg concentrations, spices, microbial growth for sporulated and non-sporulated food-borne microorganisms, marketing system

## Abstract

Ca and Mg levels were determined in five spices according to marketing system (in bulk or commercialized in glass or polyethylene terephthalate (PET) containers) and correlated with microbial growth of commonly sporulated (*Clostridium perfringens* and *Bacillus cereus*) and non-sporulated (*Listeria monocytogenes*, psychrophilic and mesophilic bacteria, and yeasts and molds) food-borne pathogens present in them, when they were previously added to the microbial culture media. The basil had the highest mean Ca and Mg level and showed the highest microbial growth in the food-borne pathogenic microorganisms studied (*p* < 0.001). For Ca, the lowest levels were measured in cloves (*p* < 0.001), which had the lowest capacity for microbial contamination. Ca and Mg contents in spices correlated linear and positively (*p* < 0.05). Ca concentrations weakly and positively correlated (*p* < 0.05) with microbial counts for almost all studied microorganisms, and Mg levels for *B. cereus*, *C. perfringens*, and mesophilic bacteria (*p* < 0.05), possibly acting as a growing factor for some sporulated and non-sporulated foodborne pathogens. These relationships are especially significant when PET vs. glass was used as a packaging material for spices.

## 1. Introduction

Spices have traditionally been used for their organoleptic properties as flavorings for sausages, meats, salads, soups, etc., as well as for their medicinal and preservative properties [1]. This fact has lead to an increase in possible industrial and home sources of sporulated and non-sporulated microorganisms in spices in recent years, due to the monitoring of inadequate hygienic-sanitary conditions during their cultivation (which are specifically affected by the hygienic-sanitary quality of the irrigation water, the fertilisers used, the specific climatic and edaphic conditions of the area, etc.), harvesting, technological treatment, storage, marketing system and final handling by consumers, which can finally lead to food-borne infections and intoxications [2,3]. For this reason, it has been indicated that it is necessary to implement good hygienic and sanitary practices throughout the supply chain of dehydrated spices, in addition to the application of validated lethal processes before distribution and retail, to ensure the necessary food safety standards [4]. In this sense, the total microbial count of revivifiable aerobic microorganisms is considered as the most commonly accepted index to know the hygienic and sanitary conditions of many foods since high counts of aerobes (such as psychrophilic and mesophilic microorganisms, and moulds and yeasts) indicate unhygienic manufacturing conditions, contaminated raw material and/or poor heat treatment [5].

Although spices are mainly sold in a dehydrated state, which ensures low microbial growth, when added to highly water-active foods that are not subjected to final heat treatment, they can be vehicles for different microorganisms that can develop rapidly, especially if consumption of these foods is not immediate [6]. On the other hand, many of these microorganisms are also capable of forming heat-resistant spores and are even able to survive in low water activity conditions, as happens with the *Bacillus* and *Clostridium* genera. Therefore, they are likely to be found in foods with medium-low water activity such as dried fruits, cereal grains, milk powders and spices [7].

For this reason, different decontamination treatments are being studied for spices such as thyme, mustard, coriander, etc., and it has been indicated that ultraviolet radiation could be employed in food preservation techniques by reducing microbial counts of aerobic mesophylls, and molds and yeasts [1]. In addition, other methods of food preservation such as heat treatments, irradiation, vacuum-packing and the use of chemical preservation agents are being used to control the development of aerobic and anaerobic pathogens responsible for foodborne diseases [8]. However, in recent years, consumers have been interested in finding more natural alternatives to control the development of pathogenic microorganisms in foods, such as spices, based on their content of essential oils [9], as opposed to the commonly used chemical preservatives [10]. In this sense, consumers are asking for less processed, safer and more stable foods, preferring to use natural ingredients [11,12], such as certain spices like cloves [3], especially to control the growth of non-sporulated anaerobic microorganisms such as *Listeria monocytogenes*, and of sporulated microorganisms such as *Bacillus cereus* and *Clostridium perfringens*. Other authors [13] have indicated that the addition of aqueous or alcoholic thyme extracts to yoghurt inhibited the development of moulds and yeasts.

On the other hand, it has been indicated that the microbial counts of non-sporulated aerobic bacteria communities and other food contaminating bacteria, present in aromatic herbs, both fresh and dehydrated, such as thyme and basil, can contribute to the estimation of consumer exposure to these microorganisms from these food sources as well as the possible health risk they might pose [14]. These researchers observed that mesophilic bacteria on basil increased in the final market products in comparison to the microbial counts in the latest field samples of this spice [14]. In fact, spices used as raw herbs are main food components that are slightly processed or even used fresh, making it easier for undesirable contaminating bacteria to reach the final consumer when previously exposed to a wide spectrum of potential contamination sources (soil, manure, irrigation water, among others) [15].

Other studies concluded that the hygienic quality of dried spices and herbs needs to be further controlled, as the production and growing conditions as well as the essential oil content vary considerably according to the region of production, the type of cultivation and the drying method used. This determines the differences found between different samples of oregano and thyme sold in Spanish markets [16]. Specifically, these findings showed that the contamination rates of aromatic herbs sold in Spain were very high (around 26%) with bacteria of the *Acinetobacter* (*A. calcoaceticus*), *Enterobacter*, *Shigella* and *Bacillus* genera being the most abundant microbial species.

According to studies conducted between 2007 and 2010, the Food and Drug Administration [17] published a report showing that more than 12% of dehydrated spices and aromatic herbs imported into the USA were contaminated with enteric pathogenic bacteria such as *Salmonella* and *Shigella* in excess of 7% of the total microbial bacterial communities. Additionally, sporulated microorganisms, such as *Bacillus spp.* (including *B. cereus*) and *C. perfringens*, among others, were also found.

In many countries, dried spices and herbs are dried in the sun and in the open air, practices that increase the probability of microbial contamination of the end product. Therefore, it is highly recommended that if these spices and herbs are part of the formulation of products that are marketed raw or ready-to-eat foods without heat treatment, they should be previously tested for total and faecal coliforms, as well as aerobic and pathogen sporulated microorganisms, such as *C. perfringens* and *B. cereus* [18].

The role of various essential elements like Ca and Mg in microbial growth has been documented mostly by in vitro experiments regarding media formulations and partially as constituents of antibacterial active packaging. On the other hand, when an antibacterial active packaging was reinforced with zinc magnesium oxide nanoparticles, an extended shelf-life of cold-smoked salmon samples overall against *L. monocytogenes* has been reported [19]. Similarly, others reported that Mg ions mitigate biofilm formation of *Bacillus* species [20], and water soluble sodium magnesium clorophyllin-gelatin films reduced *L. monocytogenes* growth in cooked frankfurters inoculated with this microorganism [21]. In the same line, it was reported that MgO and CaO powders exhibited antimicrobial activities against fungi used in the study [22]. Moreover, it has also been found that carboxymethyl chitosan/MgO composites exhibited great antimicrobial activity against *L. monocytogenes*, which could be used in food packaging [23]. Others [24] reported that Mg^2+^ and Ca^2+^, when present in high concentrations in foods, counteract the preservative effect of enterocin LR/6, limiting its antibacterial potential. Another study [25] indicated that Ca^2+^ content confers heat and hydrostatic high pressure with high heating resistance to spores of three strains of *C. botulinum*, while Mg^2+^ cations seemed to have a contrary effect. Finally, another study reported that when Ca and Mg ions were added to the medium, *B. cereus* was eradicated [26]. For *C. perfringens*, others reported that calcium nitrate suppressed sporulation and enterotoxin production [27].

Consequently, in this research we aimed to determine the Ca and Mg concentrations in five spices commonly used in Spain, namely (rosemary, oregano, thyme, basil and cloves). Additionally, we also aimed to determine whether the levels of these minerals influenced the microbial counts of commonly sporulated (*B. cereus* and *C. perfringens*) and non-sporulated (*L. monocytogenes*, mesophilic and psychrophilic and bacteria, and yeasts and molds) food-borne pathogens present in these spices. Moreover, the influence of the commercialization system used (in bulk, and polyethylene terephthalate (PET) and glass containers) on microbial counts for referred microorganisms when the dehydrated spices are mixed with the culture media and their possible correlation with present Ca and Mg concentrations will also be studied. The aim is to determine whether the content of Ca and Mg in spices can influence their potential use as preservatives of crude foods or, on the contrary, are positively related to the microbial growth of sporulated and non-sporulated microorganisms depending on the marketing system used.

## 2. Materials and Methods

### 2.1. Sampling of Dried Spices

Studied dehydrated spices (*n* = 75) commercialized in glass (*n* = 25) or in PET jars (*n* = 25), and in bulk (*n* = 25), were sampled aseptically in sterile bags in different Spanish establishments. The samples were analyzed microbiologically on the collection day. Full information on the characteristic of sampling of dried spices considered in this study has been previously reported elsewhere [3].

### 2.2. Ca and Mg Analysis in Spices

Three hundred milligrams of spice sample were mineralized by attack with HNO_3_ (66%) and HClO_4_ (60%) of supra-pure quality (Merck, Darmstadt, Germany) in a thermostatized multi-site mineralization block (Selecta, Barcelona, Spain). Mineralization was carried out in two stages according to a previous developed method. The determination of total concentrations of Ca and Mg in spices was performed by an atomic absorption spectrometer (AAS) instrument (Varian SpectraA, 140, Mulgrave, VIC, Australia). After the mineralization procedure, the samples were diluted to 10 mL with bidistillled water. Finally, additional dilution (1/500) with bidistilled water was done as a previous step to Ca and Mg measurement by flame atomic absorption spectrometry. Calibration curves were previously prepared by diluting stock solutions of 1000 mg/L in 1% HNO_3_ for the analyzed elements (Merck, Darmstadt, Germany).

The accuracy and precision of Ca and Mg measurement procedures (*n* = 10) were verified by testing the certified reference standard Apple leaves powder of the National Institute for Standards and Technology (NIST) 1515 (Gaithersburg, MD, USA). No significant differences were found between the mean element concentrations determined in this material and the certified levels (15.2 ± 0.10 and 14.9 ± 0.32, and 2.71 ± 0.12 and 2.82 ± 0.40 µg/g for Ca and Mg, respectively). Additionally, the accuracy of the methods was also tested on the basis of recovery experiments, after complete digestion of spiked spice samples with different amounts of elements from the standard solutions [19]. The calculated recoveries for each element were between 97% and 100.2% in all cases. The limits of detection (LOD) of the method for elements analyzed (87 and 16 ng/mL, for Ca and Mg, respectively) were calculated as previously reported [28]. The concentration (µg/mL) in samples was obtained by linear calibration. Every element was analyzed in triplicate in each sample of spice.

### 2.3. Microbiological Analysis Methods

In this study, we checked the microbial counts on the collection day of *B. cereus* and *C. perfringens* as sporulated bacteria, whose growth is related to poor hygienic sanitary quality. We also checked aerobic mesophilic and psychrophilic microorganisms, *L. monocytogenes*, and molds and yeast as non-sporulated microorganisms, as a vehicle for foodborne infections that can put the consumer’s health at risk.

Sample preparation (for the microbial count, 1.5 g of each of the dehydrated spices were diluted in 9 mL of sterile buffered peptone water solution and from these, decimal dilutions were made) and counting procedures for sporulated microorganisms were described in detail in another study [3] and were performed with official techniques, namely ISO 7932:2005 standard [29] for *B. cereus*, ISO 7937: 2005 standard for *C. perfringens* [30], ISO 4833:2003 standard for aerobic mesophilic and psychrophilic bacteria [31], ISO 21527-2: 2008 for molds and yeasts [32] and ISO 11290: 2017 for *L. monocytogenes* [33].

### 2.4. Statistical Analysis

The statistical study of the data was performed by the ANOVA or the Kruskal-Wallis test. Linear correlations among Ca and Mg levels and microbial counts checked in spices were also done. The significance level was set at 5% (*p* < 0.05). For data analyses, the SPSS 22.0 for Windows (IBM SPSS Inc., New York, NY, USA) program was used.

## 3. Results

Ca and Mg concentrations changed depending on the spice (Table 1, Figure 1A,B). Specifically, the mean Ca levels were statistically different among dried spices (*p* < 0.05). Specifically, the highest average Ca content was determined in basil, which was statistically higher than rosemary and oregano, which were statistically higher than thyme, and the latter was statistically higher than cloves (*p* < 0.001). For Mg, similar mean levels were determined for all spices with the exception of basil where significantly higher concentrations were measured (*p* < 0.001). The highest mean levels of Ca and Mg were determined in basil (Figure 1A,B), and the lowest in cloves but only for Ca. The packaging system (glass or PET containers) or bulk sale did not affect the Ca level in the spices (*p* > 0.05; Table 2). However, herb samples commercialized in PET presented a significantly lower mean Mg level than that found in those sold in bulk (*p* < 0.01; Table 2). This statistical significance in Mg concentration of samples sold in PET vs. bulk is probably due to dust or dirt accumulation in samples commercialized in bulk or even to differences in Mg content among the different types of spices studied.

As noted in a previous study [3], basil is the spice that presented higher microbial counts for most of the studied microorganisms. 

For the linear bivariate correlations between Ca and Mg concentrations in spices, and microbial counts of all sporulated (*B. cereus* and *C. perfringens*) and non-sporulated (mesophilic and psychrophilic bacteria, *L. monocytogenes*, and yeasts and molds) food-borne pathogens considered in the study, when aromatic herbs were previously mixed with the culture media, the Spearman’s non-parametric method was employed. Table 3 shows the values of the linear correlation coefficients (*r*) and significance levels (*p*) for these linear bivariate correlations. It can be observed that the microbial count of *B. cereus* as a sporulated microorganism and all of the non-sporulated (*L. monocytogenes*, mesophilic and psychrophilic bacteria, and yeasts and molds) food-borne pathogens considered present in spices increased significantly with the Ca concentrations (weak correlation; *r* < 0.4 with the exception of molds and yeasts for which a moderate correlation was found; *p* < 0.05; Table 3). Microbial counts of *B. cereus* and *C. perfringens* (sporulated bacteria) and of mesophilic microorganisms (non-sporulated microorganisms) rose significantly with Mg concentrations (weak correlation; *r* < 0.4 with the exception of that for *C. perfringens* for which a moderate correlation was found; *p* < 0.05; Table 3). These Ca concentrations in spices would possibly give a boosting effect to the microbial counts for sporulated and non-sporulated food pathogenic microorganisms, as it was also previously reported for other minerals [3].

The microbial count of both sporulated microorganisms (*C. perfringens* and *B. cereus*) in all of the marketing systems used, with the exception of those found for *B. cereus* when these spices were marketed in glass (*p* > 0.05), increased significantly with the Ca content measured in the spices (Table 4).

In relation to Mg levels, there was a weak positive linear correlation with *C. perfringens* growth for the three marketing systems used (bulk, PET and glass) for spices. However, the microbial count of *B. cereus* weakly but significantly increased with Mg concentrations only for spices marketed in PET (*p* < 0.05; Table 4).

In relation to non-sporulated microorganisms, the growth of *L. monocytogenes* significantly enhanced with Ca and Mg levels (as a moderate correlation for both minerals), only when aromatic spices were marketed in PET. However for the other marketing systems (in bulk and glass packaging), no linear correlation (*p* > 0.05) between the count of this microorganism and the Ca and Mg concentrations present in the spices, was found. On the other hand, the counts of psychrophilic microorganisms were linearly and positively related to Ca concentrations in spices when they were marketed in PET (as a moderate correlation) and glass (as a weak correlation), and with the Mg concentrations when marketed in bulk (as a moderate correlation). Similarly, the Ca content and growth of mesophilic microorganisms were also positively and linearly significantly correlated (*p* < 0.05) when spices were marketed in bulk and in glass (as a weak correlation for both commercialization systems). In contrast, for Mg levels, we only found a positive linear correlation with the growth of these microorganisms when the herbs were marketed in PET (as a moderate correlation). For molds and yeasts, PET appeared as the marketing system where there was again a positive and linear relationship between the measured Ca and Mg concentrations (as a moderate correlation for both minerals) and microbial growth. Additionally, mould and yeast counts also significantly increased with Mg concentrations in dried spices when marketed in bulk and glass (as a weak correlation for both commercialization systems) (*p* < 0.05). Contrarily, Ca concentrations did not increase with mould and yeast growth when spices were marketed in bulk and glass (*p* > 0.05).

## 4. Discussion

In the present work, we observed that the concentrations of Ca and Mg varied depending on the spice considered (Figure 1A,B), with significantly higher concentrations found in basil. For mean Ca content, the increasing order of cloves < thyme < oregano and rosemary < basil was observed. For Mg, the mean concentrations in decreasing order were basil > oregano, thyme, clove and rosemary.

When comparing the results found in the spices analyzed in this research work, with those determined by other authors in other countries and locations, a great variability in the levels of these minerals (Ca and Mg) was observed. Therefore, the mean concentration of Ca and Mg present in the spices is influenced by the climatic and edaphic conditions of the particular cultivation soils of the geographical area of origin, as well as the type of spice, climate and agricultural practices [40].

For Ca and Mg, the average levels determined in spices such as thyme, rosemary and oregano, in the present work in supermarkets and street vendors in South-East Spain, are considerably lower (between 20 and 40 times lower) than those indicated in the same dried aromatic herbs from local markets in Sicily (Italy) and Mahdia (Tunisia) [36] (Table 1). However, the mean Ca and Mg levels in the spices analyzed in this work are higher than those measured by other authors in other areas, as shown in the remaining studies included in Table 1.

In relation to the possible influence of the spice marketing system on the Ca and Mg concentrations present, it has been observed that the concentrations determined in spices sold in PET were lower than those sold in bulk only for Mg (Table 2). This finding as reported above is probably due to dust or dirt accumulation in samples commercialized in bulk or even to differences in Mg content among different types of studied spices.

We also estimated the daily intake of Ca and Mg as essential elements from the studied spices taking into consideration the present average concentration of 18.7 ± 0.99 and 6.20 ± 0.37 mg/g (dry weight: DW), respectively, and the mean amount consumed in Europe of 0.78 g/person/day [41]. Therefore, the average daily intake would be 14.6 ± 0.77 and 4.84 ± 0.29 mg/day (DW), respectively. Taking into account the reference dietary intakes (RDIs) for the healthy adult population (women and men aged 31–50 years) from the Institute of Medicine (IOM) for the North American population [42] set at 1000 mg/day for Ca (in men and women), and at 420 (in men) and 320 mg/day (in women) for Mg, the percentage coverage from spices of the RDIs would be 1.46 ± 0.08%, and 1.15 ± 0.07 and 1.51 ± 0.09% for Ca and Mg, respectively. As can be seen, spices are bad sources of Ca and Mg, probably because of their low intake in the daily diet, as was also indicated by other researchers [37].

Ca levels were linearly and positively related to Mg content in the studied spices (*r* = 0.447, *p* < 0.001). Previous studies indicated that Ca concentrations present in different food groups (meat and derivatives, legumes, cereals, nuts, fruits, vegetables, alcoholic and non-alcoholic beverages, dairy products, drinking water and other foods [43]) were linearly and positively correlated with those of Mg, with the exception of that present in fishery products, which reflects the existence of a positive relationship between both elements in foods, as we have additionally verified in spices.

In the present study, we observed how basil is the spice that has higher levels of Ca and Mg, as others have also found [34,36] (Table 1). In addition, for Ca levels, we found that there is a weak positive linear correlation established with the growth of *C. perfringens* (in relation to sporulated microorganisms) and all non-sporulated microorganisms studied. Mg levels were also positively and mainly weakly correlated with microbial counts of *B. cereus* and *C. perfringens* (sporulated microorganisms) and aerobic mesophilic microorganisms (non-sporulated microorganism). Consequently, basil is the spice that suffers the greatest contamination and microbial development [3] when added to the culture medium, which could be related to the fact that Ca and Mg could possibly act as growth factors for some of the determined food pathogenic sporulated and non-sporulated microorganisms, which should be checked in future studies.

For Ca, as others also found [34], the lowest levels were determined in cloves (Figure 1A), the spice in which the lowest growth of microorganisms was observed (in fact there was no microbial count of *C. perfringens* and *B. cereus* as sporulated bacteria as well as of *L. monocytogenes* among the non-sporulated ones) [3]. Likewise, the development of other non-sporulated microorganisms was lower than that established in basil [3]. Therefore, we could highlight that the lower Ca content determined in this aromatic herb (cloves) could possibly be related to the lower development of these food-borne pathogenic microorganisms, when this aromatic herb, already contaminated with them, was previously mixed with their culture media.

When relating Ca and Mg content with microbial growth and packaging systems, we found that, in PET and for both minerals, a significant positive linear correlation between all sporulated and non-sporulated microorganisms (with the exception of mesophilic microorganisms for Ca and psychrophilic microorganisms for Mg) was found. The fact that the enhancement of Ca and Mg levels significantly increased the growth of all sporulated and a large part of the non-sporulated microorganisms for spices packaged in PET could be due to the medium permeability characteristics of this plastic in relation to gases. In this sense, this characteristic would allow the creation of a passive modified atmosphere system in a natural way that balanced oxygen and carbon dioxide inside these PET containers, favouring the growth of certain microorganisms. These results are in line with that reported by other authors [44] who concluded that the effect that modified atmospheres could have on microbial growth is largely unknown, as it depends largely on the barrier properties of the materials used. In general, Gram (−) bacteria are more sensitive to CO_2_ than Gram (+) ones. However, in those containers in which a balanced atmosphere between O_2_ and CO_2_ concentrations is achieved during storage, the growth of aerobic and anaerobic microorganisms would not be inhibited in the expected way [45], as the container itself creates a passive atmosphere due to its permeability to these gases, which is in line with the findings of our study. This would explain why in our study the sporulated microorganisms (presenting forms of resistance-like spores) could survive well inside the PET containers, as did four of the non-sporulated microorganisms studied, with Ca and Mg present in spices also acting as another stimulating factor in bacterial growth, apart from the atmosphere created inside the PET containers. Our findings would also be in line with those reported by other researchers [46], who found that the sporulated microorganisms could survive well inside the polypropylene containers. These authors stated that the oxygen concentration inside the polypropylene containers intended to contain dried fruits and nuts reached equilibrium with the outside oxygen during storage, creating a modified passive atmosphere inside them, as polypropylene is a semi-permeable plastic with O_2_ and CO_2_ permeability characteristics similar to PET [46].

Foods with low water activity, such as dried spices, are susceptible to attack by moulds and yeasts that can grow in the absence of O_2_ and resist high CO_2_ concentrations. The same situation would be true for anaerobic sporulating microorganisms such as *C. perfringens*. This could explain why we found a positive linear correlation between the growth of *C. perfringens* (as anaerobic sporulating microorganism) and the Ca and Mg content in herbs as well as between the growth of moulds and yeasts (non-sporulating anaerobic microorganisms) and the Mg levels in the samples packaged in glass, since glass is an oxygen impermeable material [47]. As glass is a high barrier material, it is used for packaging foods where the transfer of gaseous matter in the form of vapours and odours from the inside of the package to the outside has to be avoided. It is known that the analysed herbs are used in traditional cooking as condiments by their odoriferous and flavouring potential due to the essential oils present in their composition, which would make the glass an ideal packaging material compared to other materials with medium barrier properties such as PET. In general, when dried spices were packaged in glass, we found that there was not a significant statistical correlation (*p* > 0.05) between Ca and Mg contents, and microbial growth for *B. cereus* (aerobic sporulating microorganism) and *L. monocytogenes* (anaerobic non-sporulating microorganism). The growth of mesophilic and psychrophilic (both non-sporulated) microorganisms in herbs was positive, linear, and significantly correlated with Ca levels in glass (*p* < 0.05), but not with Mg content (*p* > 0.05). However, the degree of growth of both sporulated and non-sporulated microorganisms was lower in herbs packaged in glass than in PET.

There is no scientific literature relating microbial growth in aromatic herbs to mineral content (Ca and Mg) and marketing systems. Therefore, with this work and a previous study carried out by our research group [3], we highlight the importance of choosing the most appropriate packaging system that would generate the least impact on the aromatic herbs during their whole life cycle, not only from the organoleptic point of view but also from that of the hygienic quality. This choice has to be related to the specific characteristics of the packaging barriers against altering gases such as O_2_ and CO_2_, and the Ca and Mg contents in herbs that could act as factors of microbial growth. Contrarily, other studies have reported that Ca and Mg added as cations or as different salts exerted an antimicrobial effect against *L. monocytogenes* [19,20], fungi [22], *Bacillus* species [21,26] and *C. perfringens* [27] when used as components of nanoparticles or active packaging, increasing the shelf-life of different foods. However, in our study the stimulating effect was performed by Ca and Mg present in spices directly on microbial growth of the food-borne pathogenic microorganisms also existing in them. Consequently, more studies should be performed in the future to discover the best packaging material for dehydrated herbs. This would deal with the growth of sporulated and non-sporulated food-borne pathogenic microorganisms in order to finally guarantee the hygienic and sanitary quality of these products and protect the consumer’s health.

## 5. Conclusions

The Ca and Mg contents varied depending on the aromatic spice type. The Ca content differed among all dehydrated spices apart from oregano and rosemary. For Mg, only average concentrations determined in basil were different. The highest mean Ca and Mg level was determined in basil. For Ca, the lowest concentrations were measured in cloves. The Ca and Mg concentrations present in spices could stimulate growth for most of the food-borne pathogenic microorganisms studied. Ca and Mg contents in spices correlated linearly and positively. The growth of sporulated microorganisms increased strongly with Ca and moderately with Mg levels present in aromatic herbs packaged in PET. For non-sporulated microorganisms, their microbial counts also increased moderately with Ca and Mg concentrations in herbs packaged in PET, with the exception of mesophilic and psychrophilic microorganisms, respectively. This relationship between Ca and Mg levels and microbial growth for most of the studied sporulated and non-sporulated food-borne microorganisms was especially significant when using PET versus glass as a packaging material for dehydrated herbs.

## Figures and Tables

**Figure 1 foods-10-01122-f001:**
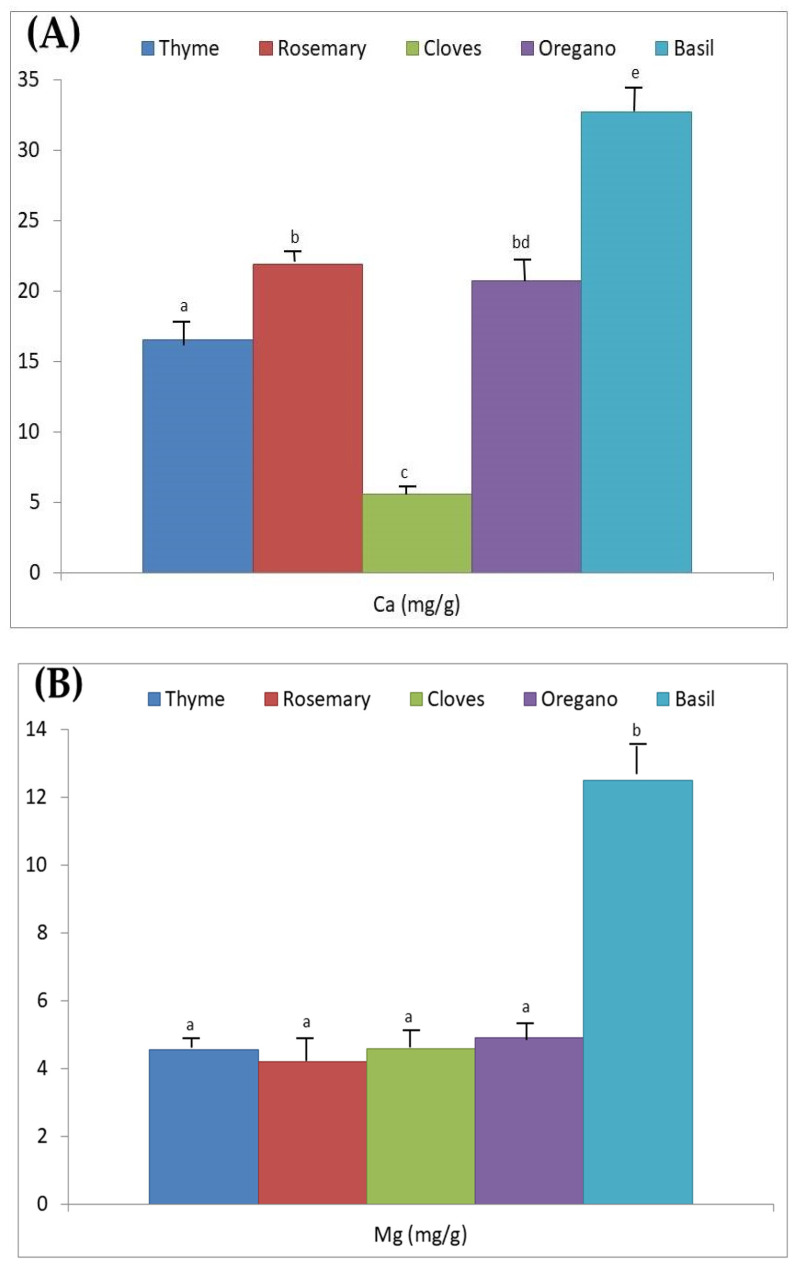
Mean (**A**) Ca and (**B**) Mg levels (mg/g) determined in dried spices. Average Ca and Mg content with different superscripts show the existence of significant differences among dried spices (*p* < 0.001).

**Table 1 foods-10-01122-t001:** Mean Ca and Mg levels (mg/g dry weight) in dried spices measured in different studies.

Spice	Ca ± SEM	Mg ± SEM	Reference
Thyme	9.58 ± 2.66	1.53 ± 0.144	[34]
Thyme	7.76 ± 6.8	2.11 ± 6.2	[35]
Thyme	3.15 ± 0.12	0.47 ± 0.01	[36]
Thyme	834 ± 70.3	380 ± 73.3	[37]
Thyme	823 ± 135	134 ± 32.2	[37]
Thyme	16.5 ± 1.21	4.55 ± 0.207	Present study
Rosemary	8.60 ± 1.91	2.41 ± 0.264	[34]
Rosemary	306 ± 43.8	36.8 ± 3.40	[37]
Rosemary	297 ± 34.5	120 ± 17.8	[37]
Rosemary	21.9 ± 0.846	4.20 ± 0.457	Present study
Cloves	6.50 ± 0.789	2.89 ± 0.122	[34]
Cloves	5.58 ± 0.315	4.59 ± 0.356	Present study
Oregano	12.7 ± 2.19	3.09 ± 1.11	[38]
Oregano	422 ± 69.5	55.0 ± 6.10	[37]
Oregano	793 ± 62.3	355 ± 20.0	[37]
Oregano	20.7 ± 1.16	4.90 ± 0.349	Present study
Basil	26.7	-	[39]
Basil	5.56 ± 1.71	0.53 ± 0.01	[36]
Basil	16.5 ± 2.96	3.13 ± 0.443	[34]
Basil	32.7 ± 1.17	12.5 ± 0.561	Present study

SEM: Standard error of the mean.

**Table 2 foods-10-01122-t002:** Mean Ca and Mg levels (mg/g dry weight) in dried spices depending on the commercialization system (in bulk, or in PET or glass containers).

Marketing System	Ca ± SEM	Mg ± SEM
PET	21.3 ± 2.18 ^a^	5.10 ± 0.494 ^a^
Glass	18.9 ± 1.91 ^a^	6.34 ± 0.787 ^a,b^
Bulk	17.3 ± 1.32 ^a^	6.53 ± 0.517 ^b^

^a,b^ Mean Ca and Mg concentrations with different superscripts expressed the existence of statistical differences (*p* < 0.01).

**Table 3 foods-10-01122-t003:** Linear correlation coefficients (*r*) and significance levels (p) between measured Ca and Mg levels (mg/g dry weight) in the dried spices and the microbial count values (CFU/g) for sporulated microorganisms (*Bacillus cereus* and *Clostridium perfringens*) and non-sporulated microorganisms (*Listeria monocytogenes*, psychrophilic and mesophilic microorganisms, and molds and yeasts) also present in spices when they were previously added to the microbial culture medium.

Microorganisms	Ca		Mg	
	*r* ^a^	*p*	*r* ^a^	*p*
Sporulated microorganisms				
*C. perfringens*	0.385	0.001	0.579	0.001
*B. cereus*	0.182	0.067	0.303	0.003
Non-sporulated microorganisms				
*L. monocytogenes*	0.265	0.007	0.198	0.054
Psychrophilic microorganisms	0.245	0.013	0.192	0.063
Mesophilic microorganisms	0.387	0.001	0.318	0.002
Molds and yeasts	0.405	0.001	0.024	0.817

^a^ Coefficient correlation is considered as a weak correlation when the *r* value is less than 0.4; a moderate correlation when the *r* value is between 0.5 and 0.7; and a strong correlation when the *r* value is higher than 0.7.

**Table 4 foods-10-01122-t004:** Linear correlation coefficients (*r*) and significance levels (*p*: between parenthesis) between measured Ca and Mg levels (mg/g dry weight) in the dried spices depending on the marketing system and the microbial count values (CFU/g) for sporulated microorganisms (*Bacillus cereus* and *Clostridium perfringens*) and non-sporulated microorganisms (*Listeria monocytogenes*, psychrophilic and mesophilic microorganisms, and molds and yeasts) also present in spices when they were previously added to the microbial culture medium.

Microorganisms	Ca			Mg		
	Bulk	PET	Glass	Bulk	PET	Glass
Sporulated microroganismis						
*C. perfringens*	0.385 ^a^ (0.012)	0.818 ^a^ (0.000)	0.724 ^a^ (0.000)	0.306 ^a^ (0.039)	0.478 ^a^ (0.028)	0.414 ^a^ (0.013)
*B. cereus*	0.345 ^a^ (0.025)	0.764 ^a^ (0.000)	0.038 (0.837)	0.174 (0.247)	0.450 ^a^ (0.041)	−0.123 (0.480)
Non-sporulated microorganisms						
*L. monocytogenes*	0.289 (0.063)	0.676 ^a^ (0.001)	−0.174 (0.341)	0.163 (0.280)	0.669 ^a^ (0.001)	−0.076 (0.683)
Psychrophilic microorganisms	0.249 (0.111)	0.654 ^a^ (0.001)	0.376 ^a^ (0.034)	0.643 ^a^ (0.000)	0.330 (0.145)	0.257 (0.136)
Mesophilic microorganisms	0.352 ^a^ (0.022)	0.132 (0.569)	0.477 ^a^ (0.006)	0.228 (0.128)	0.470 ^a^ (0.032)	0.044 (0.800)
Molds and yeasts	0.232 (0.139)	0.679 ^a^ (0.001)	−0.092 (0.617)	0.353 ^a^ (0.016)	0.685 ^a^ (0.001)	0.291 ^a^ (0.000)

^a^ Coefficient correlation is considered as a weak correlation when the *r* value is less than 0.4; a moderate correlation when the *r* value is between 0.5 and 0.7; and a strong correlation when the *r* value is higher than 0.7.

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
