# Peer review of "Ca and Mg Concentrations in Spices and Growth of Commonly Sporulated and Non-Sporulated Food-Borne Microorganisms According to Marketing Systems"

_foods, 2021, doi:10.3390/foods10051122_

Round 1
Reviewer 1 Report
The article of Garcia-Galdeano et al. concern the study of Ca and Mg concentrations in herbs/spices and their influence on the growth/sporulation of food-borne microorganisms. Given that a) spices and herbs are considered as a vehicle for pathogens to reach our food and b) the role in the microbial growth of various essential elements (Ca, Mg) in foods hasn’t been fully understood, such studies are of interest of both the fields of public health and food microbiology. However, carefully designed studies are needed to establish dose-response relations. To my opinion, this study although scientifically sound regarding the methodology (AAS estimation of Ca and Mg, microbiological examination of an adequate number of spice/dried herbs samples) suffers from misinterpretation of the results. It is one thing to observe that two variables (A and B) are correlated and it is another to conclude that A has an impact on B. Most of the times, there is another (or many other) factors that influence both of them.
In this context, I’m suggesting to the authors to revise the manuscript and address my concerns.
Introduction:
Although the subject of this study is the role of Ca and Mg in the growth and sporulation of bacteria in dried spices and herbs, there is a complete lack of background information and there are no citations regarding the role of those elements in the growth or sporulation of bacteria in general. In this manner, the introduction section failed its role i.e. to introduce the subject to the reader. So, the sentence in Line 106 “Consequently, in this research we have to determine the Ca and Mg concentrations in five spices commonly used in Spain…” seems awkward. There are many studies on the role of Ca, Mg or other elements on the growth of microorganisms either in vitro or in food models that the authors can use in order to present their motives for this study and to “match” the Ca/Mg concentrations with the bacterial population. Similarly, why did they choose to study (Lines 111-114) the influence of the container material (glass, PET, bulk) on the Ca and Mg content of spices/herbs? Are there any previous indications that Ca or Mg are leaching from the packaging material (glass or PET) in such quantities that can alter the Ca/Mg content of the product? And if so, how can you prove that those elements came from the packaging and aren’t they naturally existing in plant tissue or it isn’t just dirt or dust accumulated during the drying or handling of the herbs?
Materials and Methods:
Lines 131-133. Despite the ref [3], please provide a brief description of the AAS methodology (flame or furnace) as well as the method of the digestion of samples, the dilutions needed for the concentration estimation and the number of replications.
Lines 145-146. Please give a brief description of the methodology used particularly in dilution.
Results:
Line 168: In header of Title 1 as well as in the text. Use “mg/g DW” as unit indicator since you have studied dried material (Dry Weight) and not fresh (or Wet Weight). Additionally, table 1 could present the mean Ca and Mg content in the study not only according to the species but also according to the packaging.
Lines 166-167: Please try to be strict with the interpretation of your results. To me the results in Table 2 clearly show that there are no differences between Ca or Mg between the various commercialization systems. The statistical significance in Mg concentration of samples sold in PET or bulk is probably a) due to dust or dirt accumulation in the second or b) due to differences in the types of spices/herbs themselves. For example, what is the mean concentration of Ca or Mg if the samples are categorized according to the species AND the commercialization system?
Lines 185-185 “ It can be observed how the Ca concentrations measured in spices increased significantly with the microbial count of….”. This sentence is not correct. These are not the results of a linear regression. Table 3 doesn’t present any Ca or Mg concentrations. It is just a correlation matrix with (although significant) weak correlation coefficients (<0.75).
Lines 191, 193. Probabilities of 0.067 or 0.054 or 0.063 do not show any “tendency”. They are telling you to accept the null hypothesis (i.e. your correlation coefficient is not different than zero). Being close to the significant level of 0.05 has to do with the number of your samples. Therefore, do not try to “stretch” the meaning of your data.
Line 191. “Mg concentrations rose significantly with microbial counts of..”. Please review your sentences. Mg concentration is a depended or independent variable in your analysis? In other words, are you suggesting that Mg concentrations are influenced by the microbial population? But this is the result of a misinterpreted outcome of a correlation. You are claiming that Mg concentration increase the bacterial counts but at the same time based on your results someone else can argue that bacterial counts increase the concentration of Mg.
Line 199: “To confirm this result, future studies should be dome to add increasing quantities of Ca and also Mg, to the culture medium of the sporulated and non-sporulated microorganisms studied, to definitively check their possible stimulating effect on the growth of these microorganisms”. To confirm which result? If Ca and Mg content of culturing media (in a Petri dish) relates to the sporulation of microorganisms? I agree with this. This is a different subject of research and there are a lot of studies about it. But in your study the purpose is to present to us your evidence on how Ca and Mg content of the spices/herbs influence the microbial population or the sporulation of bacteria. Is it by leaching from spices/herbs in their culturing media? Do you know the differences on the concentration of inorganic compounds and oligo-elements (i.e. Ca and Mg) among the existing buffered peptone water (BPW) brands? BPW is the dilutant you have used according to ref [3].
Discussion:
I believe that is section is more carefully written with a few exemptions.
Lines 260-261. Is this a strange finding? The higher concentrations of elements in spices/herbs sold in bulk could it be due their exposure in environmental factors?
Lines 263-273. The estimation of the Ca and Mg daily intake from the consumption of spices/herbs is unrelated to the subject of this article.
Lines 281 -308. Overuse of self-citation [3]. Please strengthen your point with more refs.
Lines 295-298. “Therefore, we could highlight that possibly the lower Ca content determined in this aromatic herb (clove), could be related to the lower development of these food-borne pathogenic microorganisms, since cloves did not present adequate levels of Ca for adequate microbial growth”. How did you come to this conclusion? Please add some references to support this.
Lines 348-349. “There is no scientific literature relating microbial growth in aromatic herbs to mineral content (Ca and Mg) and marketing systems”. This is true. However, there is enough literature concerning the role of inorganics on the microbial growth/sporulation to use.
Based on the above, I believe that authors should partially revise their manuscript.
Author Response
29 April 2021
Manuscript ID: foods-1184931
Type of manuscript: Article
Title: Ca and Mg Concentrations in Spices and Growth of commonly sporulated and Non-Sporulated Food-Borne Microorganisms According to Marketing Systems
Authors: José María García-Galdeano, Marina Villalón-Mir, José Medina-Martínez, Sofia María Fonseca-Moor-Davie, Jessandra Gabriela Zamora-Bustillos, Lydia María Vázquez-Foronda, Ahmad Agil, Miguel Navarro-Alarcón *
First of all we would like to thank reviewer by their comments and suggestions because we consider that they really increase the scientific value of this study, as well as help to clarify some aspects of the manuscript. All changes done in the revised version of the manuscript have been highlighted in yellow in order to facilitate reviewers and editorial board their location along the text. In relation to this, we would like to make the following considerations:
1ST REVIEWER
- “The article of Garcia-Galdeano et al. concern the study of Ca and Mg concentrations in herbs/spices and their influence on the growth/sporulation of food-borne microorganisms. Given that a) spices and herbs are considered as a vehicle for pathogens to reach our food and b) the role in the microbial growth of various essential elements (Ca, Mg) in foods hasn’t been fully understood, such studies are of interest of both the fields of public health and food microbiology. However, carefully designed studies are needed to establish dose-response relations. To my opinion, this study although scientifically sound regarding the methodology (AAS estimation of Ca and Mg, microbiological examination of an adequate number of spice/dried herbs samples) suffers from misinterpretation of the results. It is one thing to observe that two variables (A and B) are correlated and it is another to conclude that A has an impact on B. Most of the times, there is another (or many other) factors that influence both of them”.
* We appreciate general comments performed by the reviewer, and therefore we have proceeded to answer and clarify the suggestions done by him/her.
- Introduction:
- Although the subject of this study is the role of Ca and Mg in the growth and sporulation of bacteria in dried spices and herbs, there is a complete lack of background information and there are no citations regarding the role of those elements in the growth or sporulation of bacteria in general. In this manner, the introduction section failed its role i.e. to introduce the subject to the reader. So, the sentence in Line 106 “Consequently, in this research we have to determine the Ca and Mg concentrations in five spices commonly used in Spain…” seems awkward. There are many studies on the role of Ca, Mg or other elements on the growth of microorganisms either in vitro or in food models that the authors can use in order to present their motives for this study and to “match” the Ca/Mg concentrations with the bacterial population.
- Following reviewer’s suggestions a new paragraph with studies relating Ca and Mg levels in food and microbial growth has been included in the revised version (lines 102-116).
- Similarly, why did they choose to study (Lines 111-114) the influence of the container material (glass, PET, bulk) on the Ca and Mg content of spices/herbs? Are there any previous indications that Ca or Mg are leaching from the packaging material (glass or PET) in such quantities that can alter the Ca/Mg content of the product? And if so, how can you prove that those elements came from the packaging and aren’t they naturally existing in plant tissue or it isn’t just dirt or dust accumulated during the drying or handling of the herbs?
- As pointed out the reviewer this was not properly reflected in the original version of the manuscript. Evidently, in this study we tried to determine if the commercialization system used for spices (in bulk, and polyethylene terephthalate and glass containers) influenced the microbial counts for studied microorganisms, and if such enhanced growths could also be additionally related with the existing Ca and Mg levels in them. Therefore these lines have been properly changed in the revised version (lines 122-125 of the revised version).
- Although the subject of this study is the role of Ca and Mg in the growth and sporulation of bacteria in dried spices and herbs, there is a complete lack of background information and there are no citations regarding the role of those elements in the growth or sporulation of bacteria in general. In this manner, the introduction section failed its role i.e. to introduce the subject to the reader. So, the sentence in Line 106 “Consequently, in this research we have to determine the Ca and Mg concentrations in five spices commonly used in Spain…” seems awkward. There are many studies on the role of Ca, Mg or other elements on the growth of microorganisms either in vitro or in food models that the authors can use in order to present their motives for this study and to “match” the Ca/Mg concentrations with the bacterial population.
- Materials and Methods:
- Lines 131-133. Despite the ref [3], please provide a brief description of the AAS methodology (flame or furnace) as well as the method of the digestion of samples, the dilutions needed for the concentration estimation and the number of replications.
- Following reviewer’s comments a more detailed explanation has been given in the revised version (lines 138-158 in the revised version).
- Lines 131-133. Despite the ref [3], please provide a brief description of the AAS methodology (flame or furnace) as well as the method of the digestion of samples, the dilutions needed for the concentration estimation and the number of replications.
- Results:
- Line 168: In header of Title 1 as well as in the text. Use “mg/g DW” as unit indicator since you have studied dried material (Dry Weight) and not fresh (or Wet Weight). Additionally, table 1 could present the mean Ca and Mg content in the study not only according to the species but also according to the packaging.
- Following reviewer suggestions “DW referring to dry weight”, has been included in the table 1 legend as well as in table 2, 3 and 4.
- Due that none of studies different than ours collected in table 1 measured Ca and Mg depending to packaging, we consider that it is not appropriate to include results of table 2 into table 1, because it could induce to misunderstanding to readers interested in this research.
- Line 168: In header of Title 1 as well as in the text. Use “mg/g DW” as unit indicator since you have studied dried material (Dry Weight) and not fresh (or Wet Weight). Additionally, table 1 could present the mean Ca and Mg content in the study not only according to the species but also according to the packaging.
- Lines 166-167: Please try to be strict with the interpretation of your results. To me the results in Table 2 clearly show that there are no differences between Ca or Mg between the various commercialization systems. The statistical significance in Mg concentration of samples sold in PET or bulk is probably a) due to dust or dirt accumulation in the second or b) due to differences in the types of spices/herbs themselves. For example, what is the mean concentration of Ca or Mg if the samples are categorized according to the species AND the commercialization system?
- We really thank reviewer’s comments and fully agree with them. Therefore corresponding changes have been done in the revised version of the manuscript (lines 187-190, revised version).
- Lines 191, 193. Probabilities of 0.067 or 0.054 or 0.063 do not show any “tendency”. They are telling you to accept the null hypothesis (i.e. your correlation coefficient is not different than zero). Being close to the significant level of 0.05 has to do with the number of your samples. Therefore, do not try to “stretch” the meaning of your data.
- Following reviewer’s comments all text related to probabilities of 067 or 0.054 or 0.063 and their relation to “data tendency” have been erased in the revised version (lines 219-221 and 224-226) .
- Following reviewer’s comments all text related to probabilities of 067 or 0.054 or 0.063 and their relation to “data tendency” have been erased in the revised version (lines 219-221 and 224-226) .
- Line 191. “Mg concentrations rose significantly with microbial counts of..”. Please review your sentences. Mg concentration is a depended or independent variable in your analysis? In other words, are you suggesting that Mg concentrations are influenced by the microbial population? But this is the result of a misinterpreted outcome of a correlation. You are claiming that Mg concentration increase the bacterial counts but at the same time based on your results someone else can argue that bacterial counts increase the concentration of Mg.
- Evidently microbial counts of B. cereus and C. perfringens (sporulated bacteria) and of mesophilic microorganisms (non-sporulated microorganisms) were those that rose significantly with Mg concentrations in spices. Therefore this sentence has been properly corrected in the revised version (lines 221-224).
-
- Line 199: “To confirm this result, future studies should be dome to add increasing quantities of Ca and also Mg, to the culture medium of the sporulated and non-sporulated microorganisms studied, to definitively check their possible stimulating effect on the growth of these microorganisms”. To confirm which result? If Ca and Mg content of culturing media (in a Petri dish) relates to the sporulation of microorganisms? I agree with this. This is a different subject of research and there are a lot of studies about it. But in your study the purpose is to present to us your evidence on how Ca and Mg content of the spices/herbs influence the microbial population or the sporulation of bacteria. Is it by leaching from spices/herbs in their culturing media? Do you know the differences on the concentration of inorganic compounds and oligo-elements (i.e. Ca and Mg) among the existing buffered peptone water (BPW) brands? BPW is the dilutant you have used according to ref [3].
- Evidently the aim of our study was to study the possible evidence on how Ca and Mg content of the spices/herbs could influence the microbial population of sporulated and non-sporulated food-borne pathogens. When spices were previously added to the culture media for studied microorganisms, they supplied both microbial contamination as well as Ca and Mg concentrations present in them, which could or could not act as a possible factor for microbial growth. We have not studied if the leaching process of Ca and Mg facilitated the transfer of these to the culture media or microorganism took directly from the spices added to it. Nevertheless, and following reviewer comments this sentence has been erased in the revised version (lines 229-232).
- Discussion:
- Lines 260-261. Is this a strange finding? The higher concentrations of elements in spices/herbs sold in bulk could it be due their exposure in environmental factors?
- This sentence has been changed as it was above done (lines 297-299). Evidently environmental factors like dust or dirt accumulation in samples commercialized in bulk could occur happen, which would justify the higher Mg levels measured in samples commercialized in bulk vs. in PET.
- Lines 260-261. Is this a strange finding? The higher concentrations of elements in spices/herbs sold in bulk could it be due their exposure in environmental factors?
- Line 199: “To confirm this result, future studies should be dome to add increasing quantities of Ca and also Mg, to the culture medium of the sporulated and non-sporulated microorganisms studied, to definitively check their possible stimulating effect on the growth of these microorganisms”. To confirm which result? If Ca and Mg content of culturing media (in a Petri dish) relates to the sporulation of microorganisms? I agree with this. This is a different subject of research and there are a lot of studies about it. But in your study the purpose is to present to us your evidence on how Ca and Mg content of the spices/herbs influence the microbial population or the sporulation of bacteria. Is it by leaching from spices/herbs in their culturing media? Do you know the differences on the concentration of inorganic compounds and oligo-elements (i.e. Ca and Mg) among the existing buffered peptone water (BPW) brands? BPW is the dilutant you have used according to ref [3].
- Lines 263-273. The estimation of the Ca and Mg daily intake from the consumption of spices/herbs is unrelated to the subject of this article.
- It is true, but anyway we consider that it supplies interesting information about Mg and Ca supply in diet from this food source which could be interesting for some researchers when tried to evaluate Ca and Mg daily dietary intakes from different food sources. In the case of spices very little studies have been performed in this issue.
- Lines 281 -308. Overuse of self-citation [3]. Please strengthen your point with more refs.
- We agree with the statement done by the reviewer. Nevertheless, we usually refer to this reference because it collects microbial growth for the 6 groups of microorganisms considered in the present manuscript. Unfortunately, after looking for additional references related with this specific subject we have not found anyone. Additional studies have been included in the revised version (lines 310, 318 and 388-391).
- We agree with the statement done by the reviewer. Nevertheless, we usually refer to this reference because it collects microbial growth for the 6 groups of microorganisms considered in the present manuscript. Unfortunately, after looking for additional references related with this specific subject we have not found anyone. Additional studies have been included in the revised version (lines 310, 318 and 388-391).
- Lines 295-298. “Therefore, we could highlight that possibly the lower Ca content determined in this aromatic herb (clove), could be related to the lower development of these food-borne pathogenic microorganisms, since cloves did not present adequate levels of Ca for adequate microbial growth”. How did you come to this conclusion? Please add some references to support this.
- In the previous study performed by our research group “García-Galdeano, et al., 2020) we found that when the cloves was added to the culture media for three of the studied microorganisms namely (B. cereus, C. Perfringens and L. monocytogenes) no microbial growth was observed. Additionally and such it can be seen in Fig. 1A, the cloves is the aromatic herb that has significantly lower Ca levels. Therefore, a possible relationship between low Ca levels and lower growth of these food-borne pathogenic microorganisms could not be discarded among other possible factors that additionally could also influence negatively the microbial growth.
- Taking into consideration statements done by reviewer this sentence has been changed in the revised version to:
- “Therefore, we could highlight that possibly the lower Ca content determined in this aromatic herb (clove), could be related to the lower development of these food-borne pathogenic microorganisms” (lines 333-335).
- “Therefore, we could highlight that possibly the lower Ca content determined in this aromatic herb (clove), could be related to the lower development of these food-borne pathogenic microorganisms” (lines 333-335).
- Lines 348-349. “There is no scientific literature relating microbial growth in aromatic herbs to mineral content (Ca and Mg) and marketing systems”. This is true. However, there is enough literature concerning the role of inorganics on the microbial growth/sporulation to use.
- It is true the statement done by the reviewer. Nevertheless, the aim of this study was not to study the influence of Ca and Mg content in the culture media on microbial growth/sporulation of the sporulated and non-sporulated food-borne pathogens studied. In fact we tried to correlate if the microbial growth of these microorganisms already existing in aromatic herbs was influenced by the commercialization system of them and if it could additionally be related with Ca and Mg levels also present in spices. When herbs were commercialized in PET vs. glass containers the different permeability to oxygen and carbon dioxide could influence in the microbial growth. In this sense we reported that “When relating Ca and Mg content with microbial growth and packaging systems, we found that in PET and for both minerals a significant positive linear correlation between all sporulated and non-sporulated microorganisms (with the exception of mesophilic microorganisms for Ca and psychrophilic microorganisms for Mg) was found”.
Reviewer 2 Report
Overall, it is a very interesting work where authors are trying to correlate the Ca and Mg in spices with the microbial growth, which can be used as a detection method to validate the microbial safety in the dried spices. I do have several comments for revision:
- Introduction section: in this section, the authors have mentioned the concerns for pathogen contamination in spices and some current practices to inactivate pathogens in spices (e.g. UV). They have also mentioned the potential use of spices as preservatives due to their content of essential oil. However, there is a link that is missing to connect this background with the aim of this study which is using Ca and Mg concentration to indicate the microbial growth in spices. Authors need to provide some linkage of this: why does Ca and Mg content matter? how do you come to this direction to measure Ca and Mg in spices? What is the current practice to detect microbial growth in spices? How can the result from this work help to increase the efficiency/benefit the industry? What is the research gap you are trying to answer?
- Line 123: should be "on the collection day".
- Line 139-149: were you testing the native microflora in the samples? or you were inoculating microbes onto the spices? if so, do you sanitize the spices before inoculation? could you provide more details on this?
- Line 180 - 238: in this section of correlation, most of the linear correlation coefficients (r) are less than 0.4. Generally, a linear correlation coefficient (r) needs to be greater than 0.7 to be considered as a strong correlation, between 0.5 and 0.7 as moderate correlation, and anything less than 0.4 is considered a weak or no correlation. If r is negative, then when it is closer to -1, the more correlated it is. Thus, it is hard to draw a conclusion based on the results presented in this study to say that Ca and Mg content is correlated with microbial growth in spices. In addition, when p>0.05, it means no significant difference. It seems no appropriate to say that "there was a tendency to a significant linear correlation" (Line 189).
Author Response
29 April 2021
Manuscript ID: foods-1184931
Type of manuscript: Article
Title: Ca and Mg Concentrations in Spices and Growth of commonly sporulated and Non-Sporulated Food-Borne Microorganisms According to Marketing Systems
Authors: José María García-Galdeano, Marina Villalón-Mir, José Medina-Martínez, Sofia María Fonseca-Moor-Davie, Jessandra Gabriela Zamora-Bustillos, Lydia María Vázquez-Foronda, Ahmad Agil, Miguel Navarro-Alarcón *
First of all we would like to thank reviewer by their comments and suggestions because we consider that they really increase the scientific value of this study, as well as help to clarify some aspects of the manuscript. All changes done in the revised version of the manuscript have been highlighted in yellow in order to facilitate reviewers and editorial board their location along the text. In relation to this, we would like to make the following considerations:
2nd REVIEWER
- “Overall, it is a very interesting work where authors are trying to correlate the Ca and Mg in spices with the microbial growth, which can be used as a detection method to validate the microbial safety in the dried spices. I do have several comments for revision”:
* First of all, we would like to thank the general comments performed by the reviewer, because they really encourage us to follow studying this difficult and interesting research area. English language and style have been fine/spell checked in the revised version. In relation to comments done by the reviewer we have to clarify the following aspects:
- Introduction section: in this section, the authors have mentioned the concerns for pathogen contamination in spices and some current practices to inactivate pathogens in spices (e.g. UV). They have also mentioned the potential use of spices as preservatives due to their content of essential oil. However, there is a link that is missing to connect this background with the aim of this study which is using Ca and Mg concentration to indicate the microbial growth in spices. Authors need to provide some linkage of this: why does Ca and Mg content matter? how do you come to this direction to measure Ca and Mg in spices? What is the current practice to detect microbial growth in spices? How can the result from this work help to increase the efficiency/benefit the industry? What is the research gap you are trying to answer?.
- The determination of Ca and Mg levels in spices matters because the possible influence on their natural content in spices on the microbial growth of microorganisms naturally existing in them, has not still been studied.
- That we know it is that is not a usual practice to evaluate the microbial contamination of spices. In other sense it has been traditionally reported that spices overall by their content in essential oils could be an alternative to the chemical preservatives. However, some of them could contrarily act as vehicles of food-borne pathogens as we have found for basil, overall when it is added to raw food no finally submitted to any heat treatment. Contrarily, the cloves would be the spice that really would control and therefore limit the microorganisms’ growth; therefore this spice could be recommended to be directly used by the industry as natural preservative for some developed food products.
- With this study we additionally try to answer if the commercialization system would influence the microbial growth for food-borne pathogens already present in spices and if the existing Ca and Mg levels in studied aromatic herbs are additionally correlated with this microbial growth.
- Line 123: should be "on the collection day".
- “On the collection day” has been included in the revised version as reviewer recommended (lines 133 and 160).
- “On the collection day” has been included in the revised version as reviewer recommended (lines 133 and 160).
- Line 139-149: were you testing the native microflora in the samples? or you were inoculating microbes onto the spices? if so, do you sanitize the spices before inoculation? could you provide more details on this?
- Evidently we were testing the native microflora in samples of aromatic herbs in order to know if they could directly be used in the industry as natural preservatives in processed foods. This has been clarified in the corrected version (lines 166-167) and included as:
- “(for the microbial count, 1.5 g of each of the dehydrated spices were diluted in 9 mL of sterile buffered peptone water solution and from these, decimal dilutions were made)”.
- “(for the microbial count, 1.5 g of each of the dehydrated spices were diluted in 9 mL of sterile buffered peptone water solution and from these, decimal dilutions were made)”.
- Evidently we were testing the native microflora in samples of aromatic herbs in order to know if they could directly be used in the industry as natural preservatives in processed foods. This has been clarified in the corrected version (lines 166-167) and included as:
- Line 180 - 238: in this section of correlation, most of the linear correlation coefficients (r) are less th*an 0.4. Generally, a linear correlation coefficient (r) needs to be greater than 0.7 to be considered as a strong correlation, between 0.5 and 0.7 as moderate correlation, and anything less than 0.4 is considered a weak or no correlation. If r is negative, then when it is closer to -1, the more correlated it is. Thus, it is hard to draw a conclusion based on the results presented in this study to say that Ca and Mg content is correlated with microbial growth in spices. In addition, when p>0.05, it means no significant difference. It seems no appropriate to say that "there was a tendency to a significant linear correlation" (Line 189)?
- Taking into consideration statements done by the reviewer several changes have been made along the manuscript, as you can check in the revised version (lines 217-219, 223-224, 242-267, 319, 321, 325, 405 and 407) as well as in the footnotes of tables 3 and 4.
- As we reported above, the sentence “there was a tendency to a significant linear correlation” has been erased in the revised version (lines 219-221 and 224-226).
Round 2
Reviewer 1 Report
I believe that the manuscript has been improved. Some last comments:
Line 140. Please start this paragraph differently. For example “ The role of various essential elements like Ca and Mg in microbial growth have been documented mostly by in vitro experiments regarding media formulations and partially as constituents of antibacterial active packaging. An extended shelf-life of cold-smoked salmon…”
Please check again if LoD of the method for elements was 8.7 and 1.6 ng/mL for Ca and Mg. I believe that such low levels of LoD are related with furnace and not with flame AAS.
Legend in Tables. Please use either (mg/g DW) or (mg/g dry weight)
Some minor errors will be corrected during the proofing process.
Author Response
9 May 2021
Manuscript ID: foods-1184931
Type of manuscript: Article
Title: Ca and Mg Concentrations in Spices and Growth of commonly sporulated and Non-Sporulated Food-Borne Microorganisms According to Marketing Systems
Authors: José María García-Galdeano, Marina Villalón-Mir, José Medina-Martínez, Sofia María Fonseca-Moor-Davie, Jessandra Gabriela Zamora-Bustillos, Lydia María Vázquez-Foronda, Ahmad Agil, Miguel Navarro-Alarcón *
Round 2
Again, first of all we would like to thank reviewer by their comments and suggestions because we consider that they really increase the scientific value of this study, as well as help to clarify some aspects of the manuscript. All changes done in the revised version of the manuscript have been highlighted in yellow in order to facilitate reviewers and editorial board their location along the text. In relation to this, we would like to make the following considerations:
1ST REVIEWER
- I believe that the manuscript has been improved. Some last comments:
- a) Line 140. Please start this paragraph differently. For example “The role of various essential elements like Ca and Mg in microbial growth have been documented mostly by in vitroexperiments regarding media formulations and partially as constituents of antibacterial active packaging. An extended shelf-life of cold-smoked salmon…”
a.1. This paragraph has been included in the 2nd revised version.
- b) Please check again if LoD of the method for elements was 8.7 and 1.6 ng/mL for Ca and Mg. I believe that such low levels of LoD are related with furnace and not with flame AAS.
b.1. The correct LOD of the method for elements is 87 and 16 ng/mL (0.087 and 0.016 mg/L, respectively) as it has been corrected in the revised version.
- c) Legend in Tables. Please use either (mg/g DW) or (mg/g dry weight)
c.1. As recommended reviewer 1, mg/g dry weight has been used in legend in tables.
- d) Some minor errors will be corrected during the proofing process.
d.1. We agree with reviewer.
Reviewer 2 Report
The authors have made revisions based upon the reviewer's comments. Here are some additional points:
- Line 101: "Please start this paragraph differently. For example", does this sentence suppose to be here?
- Line 104-106: please revise the sentence and make it more readable
- Line 120: Need to add parentheses followed "cloves".
Round 2
Again, first of all we would like to thank reviewer by their comments and suggestions because we consider that they really increase the scientific value of this study, as well as help to clarify some aspects of the manuscript. All changes done in the revised version of the manuscript have been highlighted in yellow in order to facilitate reviewers and editorial board their location along the text. In relation to this, we would like to make the following considerations:
2ST REVIEWER
- Line 101: "Please start this paragraph differently. For example", does this sentence suppose to be here?:
- a) In this paragraph, the first part of the sentence has been erased and has been changed to “The role of various essential elements like Ca and Mg in microbial growth have been documented mostly by in vitroexperiments regarding media formulations and partially as constituents of antibacterial active packaging”.
b) Line 104-106: please revise the sentence and make it more readable.
b.1. This sentence has been changed in order to do it more readable as suggested reviewer 2 to “In this sense, when an antibacterial active packaging was reinforced with zinc magnesium oxide nanoparticles, an extended shelf-life of cold-smoked salmon samples overall against L. monocytogenes has been reported”.
d) Line 120: Need to add parentheses followed "cloves".
d.1. Parentheses followed “cloves” has been added.